# Global proteomic analysis of *Cryptococcus neoformans* clinical strains reveals significant differences between latent and lethal infection

Jovany J. Betancourt,[1,2] Jason A. McAlister,[3] Jesenia M. Perez,[2,4] David B. Meya,[5] Stefani N. Thomas,[4] Jennifer Geddes-McAlister,[3] Kirsten Nielsen[1,6]

**ABSTRACT**  To predict the outcomes of disseminated fungal disease, a deeper understanding of host-pathogen interactions at the site of infection is needed to identify targets for clinical intervention and diagnostic development. *Cryptococcus neoformans* is the causative agent of cryptococcosis, the largest infectious killer of individuals living with HIV. Cryptococcal infection begins in the lungs, and loss of immunological control leads to disseminated central nervous system disease and death. Using advanced mass spectrometry-based proteomic techniques, *in vivo* infection models, and patient-derived clinical strains, we explored the proteomic profiles of *C. neoformans* infections related to differences in strain virulence. Our findings reveal that non-lethal latent infection produces a proteomic response that differs significantly from the response caused by lethal infections, and that the proteomic profiles of typical and hypervirulent infections are markedly similar despite differences in time-to-death. Overall, the mouse pulmonary proteomic response in latent infection is defined by enrichment of proteins and pathways involved in extracellular matrix organization, cell adhesion, and structural changes, while the lethal infection is dominated by host defense, translation, and metabolic processes. These results provide clinically relevant information on how infections caused by different *Cryptococcus* strains may produce significantly different outcomes. We also identified abundant fungal proteins that could be future drug targets in latent and lethal cryptococcal infection.

**IMPORTANCE**  *Cryptococcus neoformans* is a fungal pathogen that causes substantial morbidity and mortality in immunocompromised individuals. The initial infection begins in the lungs after exposure to inhaled spores, after which local immune cells respond by either killing or containing the fungal cells. Immunosuppression weakens the immune system and allows fungal cells in the lungs to escape through the circulatory system and invade the central nervous system and cause fatal disease. However, differences between fungal strains influence the severity of disease manifestation. Our group has previously described genetic differences that contribute to strain-specific disease manifestations. In this study, we expanded our analysis to investigate the proteomic differences between strains of *C. neoformans* to identify candidate proteins and pathways that contribute to disease manifestation. We found that latent infection differs significantly from lethal disease from both the host and pathogen proteomic perspectives and identified several fungal protein targets for future study.

**KEYWORDS**  *Cryptococcus neoformans*, proteomics, clinical isolates, immune mechanisms, lung defense, lung infection

**Peer Reviewer** Camaron R. Hole, The University of Tennessee Health Science Center, Memphis, Tennessee, USA

Address correspondence to Kirsten Nielsen, kirstennielsen@vt.edu.

The authors declare no conflict of interest.

The outcome of disseminated fungal disease is often the result of a combination of host defenses and pathogen offenses. Through advancements in proteomic technology, we are better able to investigate these complex interactions to identify how host-pathogen warfare is waged and to better predict disease outcomes. *Cryptococcus neoformans* is an opportunistic fungal pathogen that accounts for up to 20% of the global HIV-related mortality (1, 2). Exposure to *C. neoformans* occurs through inhalation of spores from the environment, with disseminated disease (cryptococcosis) occurring as a result of immunosuppression and failure to contain the infection to the lungs. While early and aggressive clinical treatment of the infection and management of the underlying human immune deficits is crucial for improving survival, studies have shown that pathogen-specific differences in virulence factors may further contribute to disease outcomes (3, 4). Indeed, infecting immunocompetent A/J mice with patient-derived *C. neoformans* clinical strains recapitulates the disease outcomes of the original human host (5), and the Cryptococcus mouse model has provided insight into pathogen virulence determinants and host defense strategies (6, 7). Therefore, it is of immense clinical relevance to better understand how differences in infecting strains impact disease outcomes. In pursuit of this, our group previously conducted a genome-wide association study to identify single-nucleotide polymorphisms contributing to human disease (8) and strain-specific virulence differences in *C. neoformans* clinical strains (3). To understand how protein differences among *C. neoformans* clinical strains influence strain virulence and disease outcomes, we analyzed the proteomic profiles of clinical strains of varying virulence.

Mass spectrometry-based proteomic profiling of patient-derived pathogens has previously been successful in identifying differentially enriched proteins and virulence factors (9–16). Moreover, proteomic profiling of infected host tissue can reveal which proteins and pathways are involved during microbial defense, providing insight into what goes right or wrong during the disease response (12, 17). An effective approach to investigating the proteome of infection is to process host and pathogen cells from infected tissue together and then bioinformatically separate the organisms' proteomes during analysis. This dual perspective approach can be utilized to probe the entire protein landscape of host-pathogen interactions and identify relevant proteins for clinical intervention or diagnostic identification during the same experiment. Here, we used global proteomic profiling to investigate the proteomic changes in murine lungs and *C. neoformans* clinical strains of varying virulence. We show that latent infection induces a significantly different host immune response than lethal infection with either typical or hypervirulent strains. Our findings reveal how the non-lethal latent infection induces a proteomic response that differs significantly from the response caused by lethal infections, and that the proteomic profiles of typical and hypervirulent infections are unexpectedly similar despite differences in time-to-death. These results provide clinically relevant information on how infections caused by different *C. neoformans* strains may produce significantly different outcomes. We also identified candidate target fungal proteins with higher abundance in latent and lethal cryptococcal infection.

## MATERIALS AND METHODS

### *Cryptococcus* culture growth conditions

*C. neoformans* serotype A clinical isolates UgCl223 (18) (latent) and UgCl422 (18) (hypervirulent), along with the typical laboratory reference strain KN99α (19), were streaked from 30% glycerol stocks onto plates containing yeast extract peptone dextrose (YPD) and chloramphenicol agar and incubated at 30°C for 48 h. A small amount of culture was then transferred to 2 mL of YPD broth and incubated for 18 h at 30°C and at a shaking speed of 225 RPM. Culture samples (*n* = 4 per strain) were washed and resuspended in 1 mL PBS before being snap frozen in liquid nitrogen and stored at −80°C.

## Mouse infections

A/J (Strain #000646, Jackson Laboratories, Bar Harbor, ME) mice were infected intranasally with $5 \times 10^4$ cells of UgCl223 to generate latent infections (7), or with KN99α or UgCl422 to generate typical lethal or hypervirulent lethal infections, respectively ($n =$ 35; five uninfected, 10 per infection) (3, 5). After 14 days post-infection (DPI), mice were euthanized by $CO_2$, and their lungs were excised and collected into 10 mL tubes. Tissue samples were then snap-frozen in liquid nitrogen and stored at −80°C.

## Sample preparation

Lung and culture samples were processed as previously described (12, 17). Briefly, the samples were physically homogenized via bead beating for at least 5 min in 100 mM Tris (pH 8.4) buffer containing protease inhibitor cocktail (Roche, Basel, Switzerland). Sodium dodecyl sulfate (SDS) at 20% was added to a final concentration of 10% (vol/vol), and the samples were physically lysed by five cycles of probe sonication (30% amplitude, 30 s on, and 30 s off). Samples were reduced by 1% (vol/vol) of 1 M dithiothreitol and incubated on a shaking heating block at 95°C and 800 RPM for 10 min. Samples were then alkylated with 10% (vol/vol) of 0.55 M iodoacetamide, incubated in the dark at room temperature for 20 min, centrifuged to remove debris, diluted with 100% ice-cold acetone (80% [vol/vol]), and incubated at −20°C overnight. Protein pellets were collected by centrifugation at 10,000 × $g$ and 4°C, washed with 80% ice-cold acetone, and air dried. Protein pellets were resuspended in 200 µL of 8 M urea, 40 mM HEPES buffer, and sonicated in a water bath at 4°C. Protein concentration was quantified via bovine serum albumin (BSA) assay. Protein samples were diluted to 2 M urea in 600 µL of 50 mM ammonium bicarbonate. Twenty-five micrograms of protein were transferred to new LoBind tubes (Eppendorf, Hamburg, Germany), digested with Pierce trypsin/Lys-C protease cocktail (protein:enzyme ratio of 25:1, Thermo Scientific, Waltham, MA), and incubated overnight at room temperature. Digestion was stopped with 10% (vol/vol) stopping solution (20% acetonitrile and 6% trifluoroacetic acid). Peptides were isolated using C18 triple-layer resin Stop And Go Extraction (STAGE) tips, dried down using a speed vacuum for 45 min at 45°C, and stored at 4°C.

## Liquid chromatography-tandem mass spectrometry

Liquid chromatography-tandem mass spectrometry (LC-MS/MS) was performed as described previously (20). Briefly, digested peptide samples were resuspended in 0.1% formic acid and injected into an Orbitrap Exploris 240 hybrid quadrupole-Orbitrap mass spectrometer, coupled to a Vanquish Neo ultra-high-performance liquid chromatography system (Thermo Fisher Scientific, Waltham, MA). Peptides were separated using an in-line 75 mm × 50 cm PepMap RSLC EASY-Spray column packed with 2 mm C18 reverse-phase silica beads (Thermo Fisher Scientific, Waltham, MA). Peptides were electrospray ionized into the mass spectrometer with a linear gradient of 4% to 45% mobile phase B, containing 80% acetonitrile and 0.1% formic acid (mobile phase A 0.1% formic acid in water) over a 3-hour gradient at 300 nL/min, followed by a 15-minute wash at 100% mobile phase B. MS data were acquired in the Orbitrap mass analyzer by data-independent acquisition (DIA) with MS1 at 400–900 m/z and resolution of 60,000, and MS2 scans of 400 to 900 m/z at intervals of 10 m/z and a resolution of 15,000.

## Data processing

Raw data processing and peptide library search for mouse proteins were performed using MaxQuant software (Ver. 2.6.4.0, Max Planck Institute of Biochemistry, Planegg, Germany) (21). MaxDIA was enabled and searched against a *Mus musculus* UniProt proteome reference library (Taxon ID 10090, 87492 entries, downloaded March 4th, 2025). The digestion parameter was set to trypsin/P, label-free quantification was enabled, and the minimum ratio count was set to one. The minimum number of peptides to identify a protein was set to two. Matching between runs was enabled.

For the identification of *C. neoformans* proteins, data processing and a non-library search were performed using DIA-NN software (Ver. 1.9.2, Ralser Group, University of Cambridge, Cambridge, England) (22) and the *C. neoformans* serotype A H99 UniProt reference proteome (Taxon ID 235443, 7439 entries, downloaded September 8th, 2024).

The raw LC-MS/MS and relevant files were deposited to the ProteomeXchange Consortium via the PRIDE (23) partner repository with the data set identifier PXD063417 (Project accession: PXD063417, Token: 9DZ3u5h8R6g7).

## Statistical analysis

Statistical analysis was performed using Perseus software (Ver. 2.1.3.0, Max Planck Institute of Biochemistry, Planegg, Germany). Samples were annotated by infection condition. Then, LFQ intensities were $\log_2$ transformed and filtered for valid values detected in ≥60% of samples from one condition. Missing values (NA) were imputed by singular value decomposition (SVD) or maximum likelihood estimation (MLE), and intensities were normalized by median subtraction. Statistical significance (*P*-value < 0.05) was determined using Student's *t* test with Benjamini-Hochberg multiple hypothesis correction at an FDR of 0.05. The fudge factor (s0) for each comparison was tuned using the siggenes R package (fudge2()) (24). Principal component analysis (PCA) clusters were generated using a custom Python K-means clustering script (*K* = 3). The number of *K* clusters was determined using the elbow method. Heat maps and hierarchical dendrograms of protein abundance were made using heatmapper.ca (Wishart Research Group, University of Alberta, Edmonton, Canada) (25). Venn diagrams were adapted from DeepVenn (Hulsen Group, Philips, Eindhoven, Netherlands) (26).

## RESULTS

### Latent cryptococcal infection induces a significantly different host pulmonary proteomic response from lethal infection

We previously showed that the histological and immunological characteristics of pulmonary cryptococcal infection differ depending on the infecting isolate (3). To determine how strain-specific virulence differences influence the host's pulmonary proteomic response, we infected mice with the latent disease clinical isolate UgCl223 (7), the typical disease reference isolate KN99α, or the hypervirulent disease clinical isolate UgCl422 (3, 5), and characterized the changes in global host pulmonary proteomes at 14 days post-infection (DPI) compared to uninfected mice.

First, we performed comparisons between infected and uninfected mouse proteomes (Fig. S1) to identify significantly altered protein abundances (Table S1). We then generated a principal component analysis (PCA) plot and observed that latent-infected mice were separated from typical-infected, hypervirulent-infected, and uninfected mice (Fig. 1A). We applied a K-means clustering algorithm and found that Cluster three contained all the latent infection proteomes (Fig. 1A). A Chi-squared analysis of cluster membership showed that sample distribution was non-random (*P*-value < 0.001). A Euclidean-clustered heat map of all significant proteins revealed that typical- and hypervirulent-infected mice clustered together, while latent-infected mice clustered separately (Fig. 1B). Interestingly, the uninfected mouse proteomes clustered closer to the typical- and hypervirulent-infected mice than to the latent-infected mice. We compared protein hits from infections to uninfected mice and identified 781 significantly altered proteins (Fig. 1C). A typical infection exhibited 207 significantly different protein abundances compared to uninfected: 74 lesser abundant and 133 greater abundant proteins. Latent infection exhibited 600 significantly different protein abundances: 443 lesser abundant and 157 greater abundant proteins. Hypervirulent infection exhibited 89 significantly different protein abundances: 17 lesser abundant and 72 greater abundant proteins. We observed more overlap between typical and hypervirulent infections (58 enriched proteins), while latent infections compared to typical and hypervirulent comparisons had less host protein overlap (19 and 11 enriched proteins, respectively).

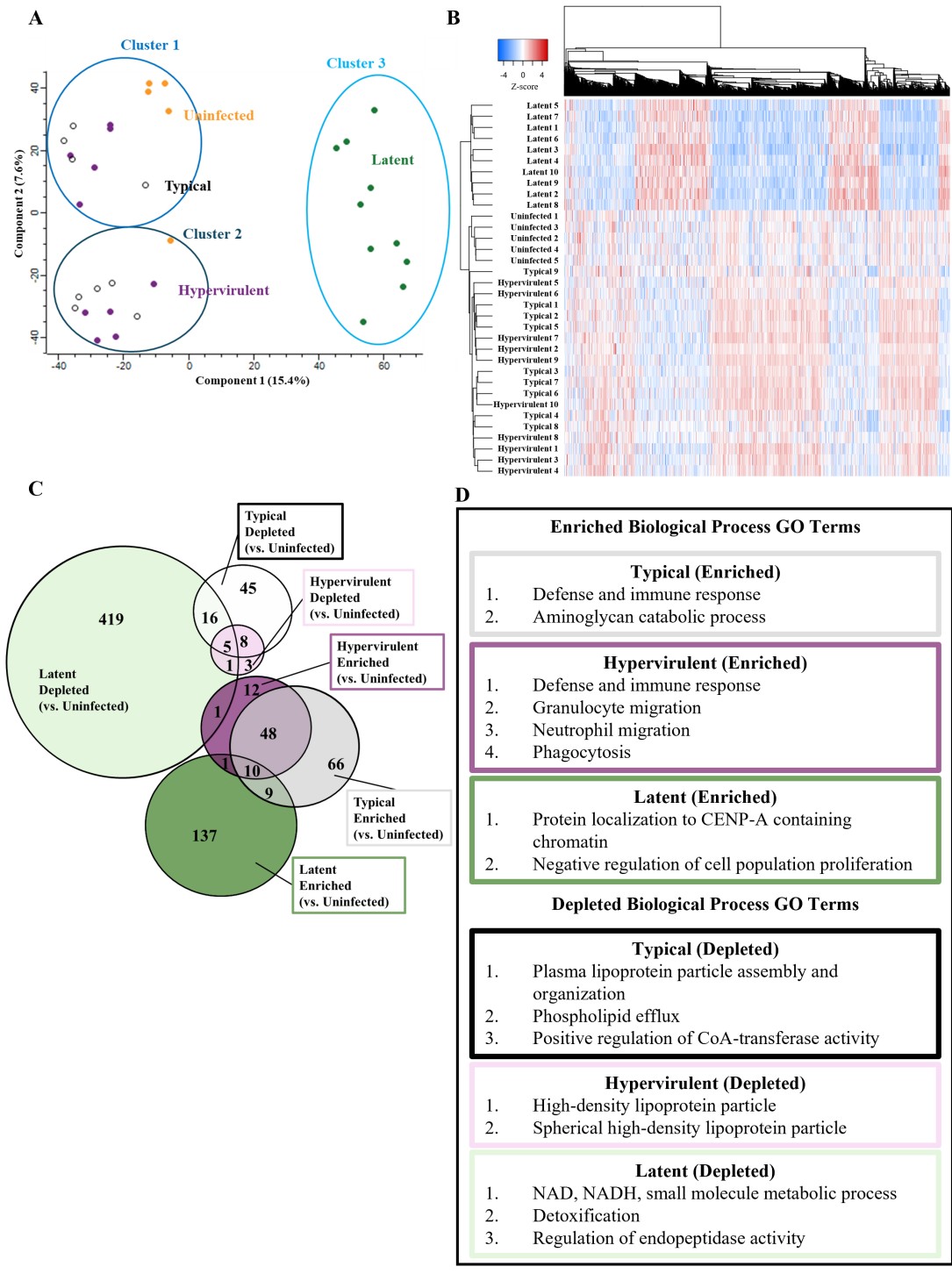

**FIG 1** The mouse lung proteome responds differently to latent cryptococcal infection than to lethal infections. Global proteomic analysis of mouse lungs infected with typical (KN99α), latent (UgCl223), or hypervirulent (UgCl422) *C. neoformans* strains at 14 days post-infection or uninfected was performed. (A) Principal component analysis (PCA) of typical (black), latent (green), hypervirulent (purple), or uninfected (orange) mouse lung proteomes. Three clusters were identified using a K-means clustering algorithm and a Chi-squared analysis (*P*-value = 0.00000221). (B) Euclidean-clustered heat map of all significant protein hits. Dendrograms of protein IDs and sample proteomes show substantial grouping of proteins in a disease-dependent manner. (C) Venn diagram depicting the number of overlapping or unique depleted (infected had less than uninfected) or enriched (infected had more than uninfected) significant proteins. (D) STRING pathway analysis showing the significantly enriched biological process gene ontology (GO) terms for each comparison (FDR < 0.05).

Interestingly, we observed only a single protein overlap between depleted latent and enriched hypervirulent conditions: the atypical pro-inflammatory glutathione S-transferase protein Gsto1 (27).

The significantly enriched and depleted proteins were then entered into the STRING database to determine the enriched pathways based on their biological process gene ontology (GO) terms (Fig. 1D). Compared to uninfected mice, mice infected with the typical isolate showed enrichment in pathways involved in defense and immune responses, as well as aminoglycan catabolic processes, and exhibited depletion in lipoprotein particle assembly and organization, phospholipid efflux, and regulation of CoA-transferases. Mice infected with hypervirulent infection also had enrichment of pathways involved in defense and immune response, along with granulocyte and neutrophil migration and phagocytosis, while similarly exhibiting depletion in lipoprotein particle organization. Latently infected mice had enrichment in pathways involved in protein localization to CENP-A-containing chromatin and negative regulation of cell population proliferation and depletion in NAD, NADH, small molecule metabolism, detoxification, and endopeptidase regulation.

Infections were also compared to each other to identify disease manifestation-specific differences (Fig. 2). Compared to the typical and hypervirulent infections, latent infection was enriched for extracellular matrix organization, cell-substrate adhesion, circulatory system processes, and cellular responses to growth factor beta, while both typical and hypervirulent infections were enriched in small molecule metabolism, biosynthetic processes, and translational pathways (Fig. 2A and B). Typical and hypervirulent infections did not have significantly different pathway enrichment, but did differ significantly in one protein: the pro-type 2 arginase protein Arg1 (28) (Fig. 2C).

We performed a direct analysis of protein abundances from the identified GO term pathways. We analyzed 12 proteins from the defense response pathway: Arg1, Cd68, Epx, Ifgr1, Mmp12, Mpo, Muc5b, Ear10, Gtso1, B4galt1, Irgm2, and Prtn3 (Fig. 3); three aminoglycan catabolism proteins: Chil1/Chia1, Chil3, and Chil4 (Fig. 4); and eight extracellular matrix proteins: Col1a1, Col1a2, Col3a1, Col4a4, Col5a2, Fga, Fgg, and Fgb (Fig. 5). The lethal typical and hypervirulent infections had significantly higher abundances of defense response proteins, except the interferon gamma receptor protein Ifngr1, which was significantly increased in latent infection (Fig. 3). Moreover, Gsto1 was significantly depleted in latent infection. Typical infection produced significantly higher abundances of Arg1, the pro-type 2 eosinophil-derived protein Epx (29, 30), the immunomodulatory metalloproteinase Mmp12 (31, 32), and the negative regulator of apoptosis B4galt1 (33, 34), while hypervirulent infection increased the levels of Irgm2 and Prtn3. All *C. neoformans* infections induced significant increases in the aminoglycan catabolism proteins but were higher in lethal infections (Fig. 4). A typical infection produced even more significantly elevated Chil1/Chia1 and Chil3. Latent infection exhibited significant increases in collagens (Col 1a1, Col1a2, Col3a1, Col4a4, and Col5a2) and fibrinogens (Fga, Fgg, and Fgb) (Fig. 5). Interestingly, typical infection resulted in significant depletion of Col1a1 and Fgb.

We previously showed that latent cryptococcal infection produces lung granulomas (6). Thus, we also examined five additional host proteins associated with granuloma formation: the reactive oxygen species (ROS)-generating proteins CYBB and Ncf1, the immune cell alarmin protein S100a9, the extracellular matrix component Vtn, and the lung protective apolipoprotein ApoE (35–38) (Fig. 6). Granuloma formation is associated with deficits in ROS generation (36, 39), increases in extracellular matrix organization (35), and promotion of neutrophil and monocyte migration (38). Latent infection produced significantly lower CYBB and Ncf1 compared to typical and hypervirulent infections, and also significantly lower S100a9 than lethal infection and uninfected. Additionally, Vtn abundance was significantly lower in hypervirulent infection, and ApoE abundance was low in latent infection.

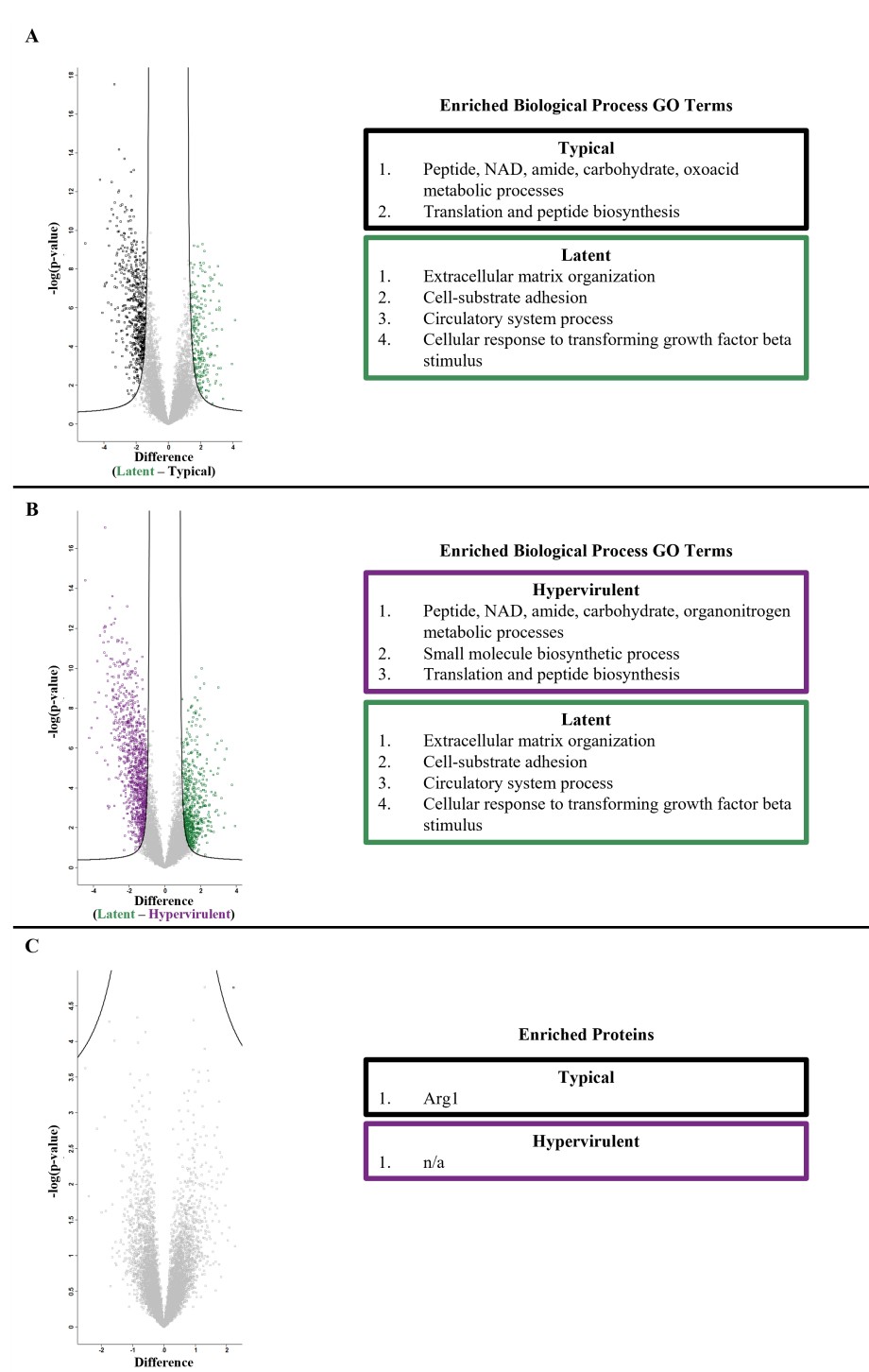

**FIG 2** Typical and hypervirulent strains induce similar proteomic profiles compared to latent infection. Volcano plots were produced using Perseus from $t$-test comparisons of mouse lung proteome profiles at 14 days post-infection from (A) latent versus typical infection, (B) latent versus hypervirulent infection, or (C) typical versus hypervirulent infection. Enriched biological process GO terms from significant proteins for (A) and (B) were determined. There were three significant proteins identified in (C) (Arg1, Arnt, and Irgm2). Fudge factor (s0) was tuned using the siggenes R package (fudge2()) for each comparison (A s0 = 1.23, B s0 = 1.23, C s0 = 0.2). Significance was determined at an FDR < 0.05.

## Clinical strains are more similar to each other than to the reference strain *in vitro*

To determine whether there were intrinsic differences in the proteomes of the clinical isolates that may contribute to their observed differences in disease manifestation, we performed global proteomic analysis and comparisons of *C. neoformans* cultures grown in the rich medium YPD *in vitro* (Fig. S2; Table S2).

We performed PCA and found that component one represented differences between two latent culture samples and the rest of the cultures (24.3%) and component two separated the clinical isolates (latent and hypervirulent) from the typical KN99α cultures (20.1%) (Fig. 7A). Applying K-means clustering showed that Cluster 1 contained all four hypervirulent culture samples and two latent culture samples; Cluster 2 contained all four typical culture samples; and Cluster 3 contained two latent culture samples. Chi-squared analysis for random distribution indicated significant non-random clustering (*P*-value < 0.01). We visualized the relationship between cultures and significant protein abundance and observed closer clustering of latent and hypervirulent cultures together (shared the same clade) compared to typical cultures (isolated clade) (Fig. 7B). We noted 25 overlapping proteins between latent and hypervirulent cultures compared to eight overlapping proteins in typical and hypervirulent cultures and eight overlapping proteins in typical and latent cultures (Fig. 7C). We then sought to determine pathway enrichment using Markov Cluster Algorithm (MCL) clustering to identify enriched protein function annotations (Fig. 7D). In general, typical cultures were enriched for terms associated with energy derivation by oxidation, sugar metabolism, DNA replication, choline metabolism, mitochondrial intermembrane space, and cytosolic large ribosomal subunits; latent cultures were enriched for DNA damage response and repair, galactose and polysaccharide metabolism, DNA metabolism, and mixed biosynthetic processes;

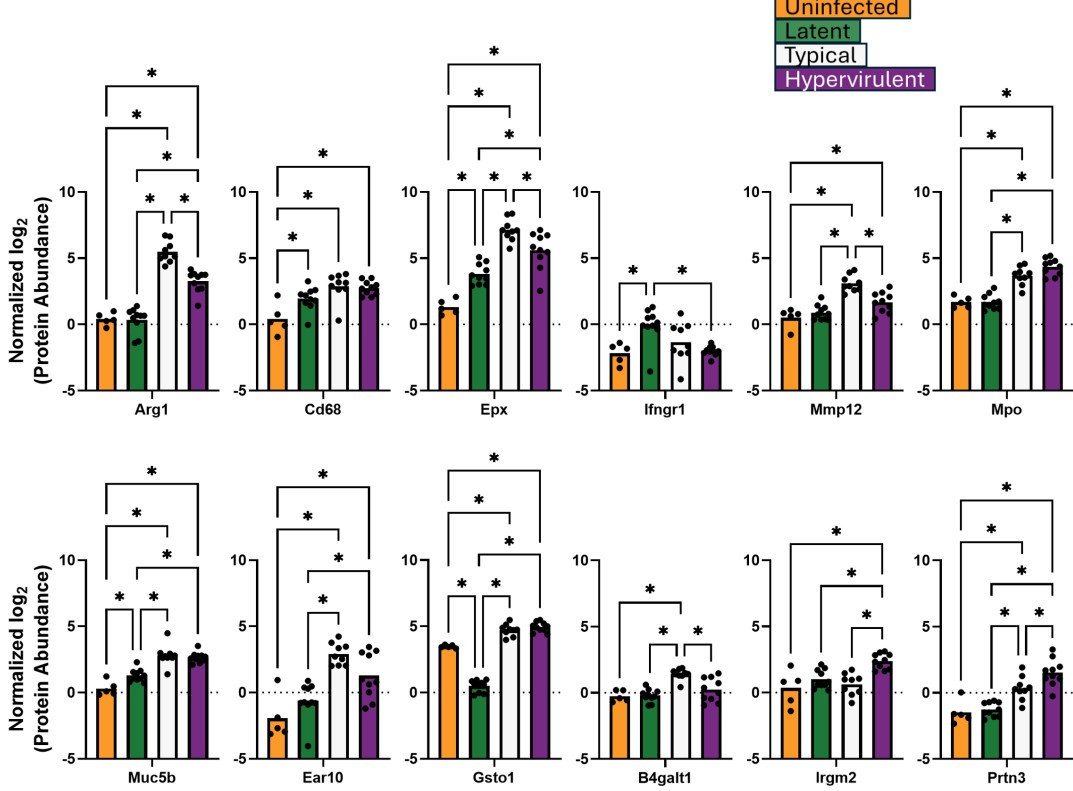

**FIG 3** Host defense response proteins were induced during latent infection. Abundances of lung proteins involved in defense response were analyzed by one-way ANOVA with Tukey's post-hoc correction. *P*-value <0.05.

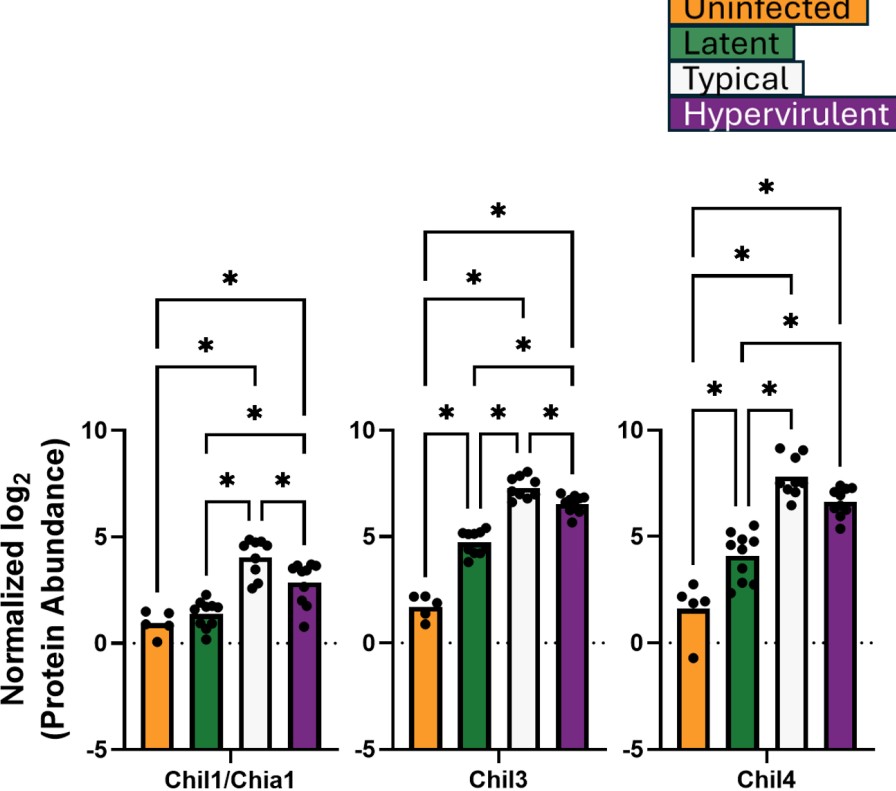

**FIG 4** Aminoglycan catabolism protein responses were highest in typical infections. Abundances of lung proteins involved in aminoglycan catabolism were analyzed by one-way ANOVA with Tukey's post-hoc correction. *$P$-value <0.05.

and hypervirulent cultures were enriched for uncharacterized endoplasmic reticulum proteins and mixed sulfur carrier and L-phenylalanine catabolism. Latent and hypervirulent cultures were not significantly different enough to produce MCL clusters.

## Latent *C. neoformans* strain produces a unique proteome compared to lethal disease-causing strains

While our findings showed that our latent and hypervirulent clinical strain *in vitro* culture proteomes are more similar to each other than our typical reference strain, we sought to determine whether *in vivo* proteomic differences may contribute to disease manifestation-specific pathogenesis. Thus, we performed global proteomic analysis of *C. neoformans* proteins isolated from infected mouse lungs at 14 DPI, compared *in vivo* proteomes (Fig. S3), and identified significantly altered proteins (Table S3).

From our PCA plot analysis, we observed variability between lethal and latent strains along component 1 (23.6%) and separation of the latent strain from the lethal strains along component 2 (18.0%) (Fig. 8A). After applying our K-means clusters, Cluster 1 contained eight latent and one typical proteome. Cluster 2 contained five typical and six hypervirulent proteomes, and Cluster 3 contained three typical, four hypervirulent, and two latent proteomes. A Chi-squared analysis showed that cluster membership was significant for non-random distribution ($P$-value < 0.001). We found that the latent *C. neoformans* strain grouped separately from typical and hypervirulent strains based on abundance of significantly altered protein hits (Fig. 8B). Our comparisons between strains revealed that our typical strain shared all seven enriched proteins with our hypervirulent strain, while the hypervirulent strain exhibited one unique protein, the DNA replication

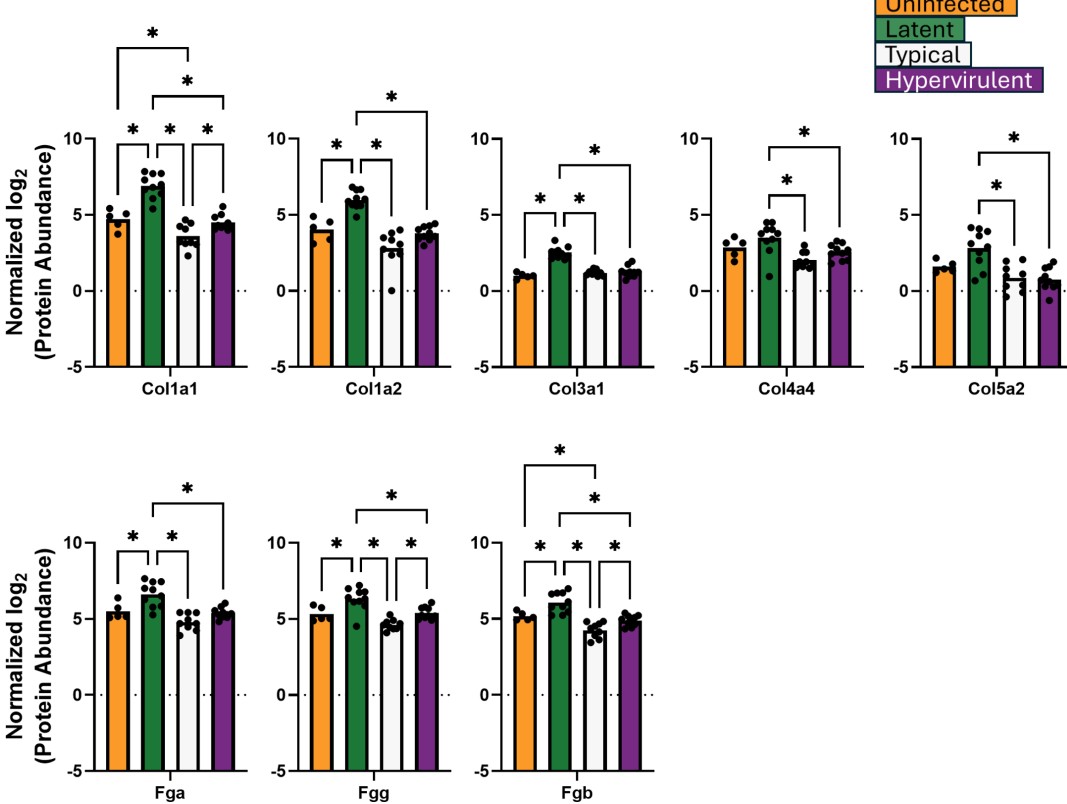

**FIG 5** Extracellular matrix organization was induced during latent infection. Abundances of lung proteins involved in extracellular matrix organization were analyzed by one-way ANOVA with Tukey's post-hoc correction. *$P$-value <0.05.

licensing factor Mcm3 (CNAG_00099) (Fig. 8C). We also showed that while six enriched latent proteins were shared between comparisons, five and eight proteins were specific to comparisons of the latent strain with typical and hypervirulent strains, respectively. To increase statistical power for our protein pathway enrichment analysis, we grouped all the significant latent proteins under the "latent" category and the typical and hypervirulent proteins under the "lethal" category. A STRING database search of enriched GO terms revealed significant enrichment of translation, RNA binding, GTP binding, and nucleoside-triphosphate and ribonucleotide activity molecular function pathways for lethal strains, and chromatin structure, protein heterodimerization, GTPase activity, and binding pathways for the latent strain (Fig. 8D).

We further categorized our significant protein identifications into two groups: nucleosome core-related proteins (Fig. 9), which include histone H2A, histone H3, histone H4, pre-mRNA splicing factor Rse1, H/ACA ribonucleoprotein complex subunit Cbf5, vacuolar protein-sorting-associated protein 4, Rab family protein, and GTP-binding protein RYH1; and translation-related proteins (Fig. 10), which include S-adenosyl-methionine synthase, elongation factor 1-alpha, elongation factor 2, elongation factor Tu, calmodulin, Hsp90-like protein, small monomeric GTPase, serine/threonine-protein phosphatase, mitochondrial aconitate hydratase, peptide chain release factor 1, ATP synthase subunit alpha, and ATP synthase subunit beta. For the nucleosome core proteins, lethal strains produced more vacuolar protein-sorting-associated protein 4 (Fig. 9). The latent strain produced significantly higher levels of DNA replication licensing factor H/ACA ribonucleoprotein complex subunit Cbf5, histone H2A, pre-mRNA splicing factor Rse1, histone H3, histone H4, GTP-binding protein RYH1, and Rab family protein (Fig. 9). For the translation-related proteins, the lethal strains exhibited significantly higher abundances of S-adenosylmethionine synthase, elongation factor 1-alpha, elongation factor 2, peptide chain release factor 1, and the stress-response Hsp90-like

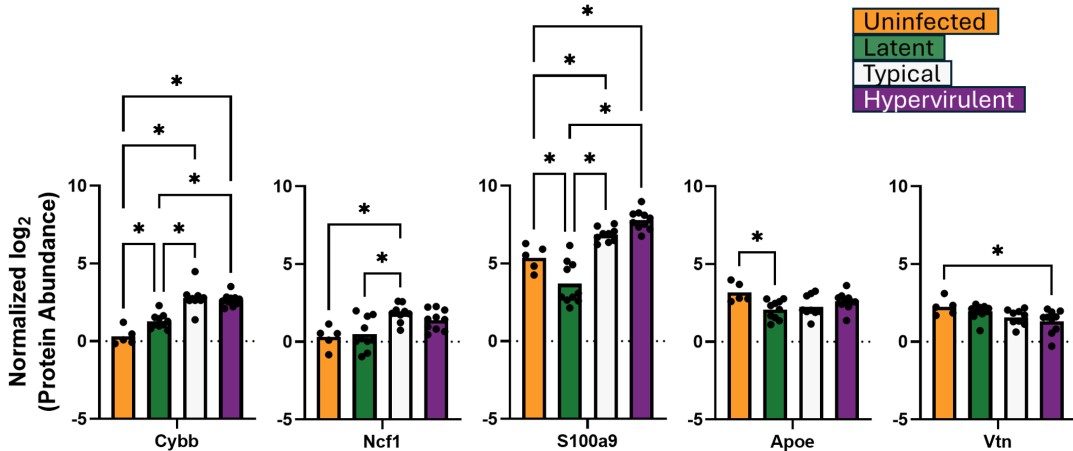

**FIG 6** Latent infection induces protein responses associated with reactive oxygen species (ROS)-deficient granuloma formation. Abundances of lung proteins involved in granuloma formation were analyzed by one-way ANOVA with Tukey's post-hoc correction. *$P$-value <0.05.

protein (Fig. 10). The latent strain exhibited increased elongation factor Tu, mitochondrial aconitate hydratase, ATP synthase subunit beta, ATP synthase subunit alpha, small monomeric GTPase, serine/threonine-protein phosphatase, and calmodulin (Fig. 10).

## DISCUSSION

We sought to understand the proteomic landscape of cryptococcal lung infection and identify the protein and pathway differences that contribute to host defense responses and fungal virulence phenotypes. We found that latent cryptococcal infection is markedly different from lethal infection from both the host and pathogen perspectives of the host-pathogen dynamic. We identified several proteins and pathways that were associated with the observed differences in disease manifestation (Fig. 11). We report that (1) the host response to latent infection involves enrichment of extracellular matrix organization pathways and interferon gamma (IFNγ) receptor protein Ifngr1; (2) the host response to typical and hypervirulent infections is enriched for defense response, biosynthetic, and translational processes; and (3) the *in vivo* cryptococcal proteomes of the latent strain UgCl223 differ significantly from those typical KN99α and hypervirulent UgCl422 lethal strains, with the proteomes of the lethal strains not differing significantly from each other. Additionally, we observed that *C. neoformans* clinical strains are more similar to each other *in vitro* than to the lab-generated reference strain KN99α. Ultimately, our results show that there are several key, clinically relevant disease manifestation-specific proteins and pathways that differ and provide insight into the mechanisms underlying cryptococcal disease outcomes.

Our analysis of the global host proteome of cryptococcal infection showed that latent, typical, and hypervirulent infection produced unique protein signatures that significantly separated the latent response from the lethal responses. While cryptococcal infection predictably elevated several defense response proteins, differences in host protein abundances between the infections suggest stark divergence in overall host response. Previous studies established that *C. neoformans* KN99α typically produces a non-protective type 2-dominant immune response characterized by eosinophilia, IL-4 and IL-13 production, and type 2 helper T cells (3, 6, 7, 30, 40–45). In agreement, we show here that the KN99α typical infection produced the greatest elevation in type 2-related proteins, such as Arg1, Epx, and Ear10, while latent infection produced the least.

We also identified that the aminoglycan catabolism pathway, containing chitinase-like proteins (CLPs) Chil1/Chia1, Chil3, and Chil4, is enriched in latent and lethal infection. Chil1/Chia1 was most abundant in lethal infections. CLPs induce proinflammatory type 2 immune responses and tissue remodeling when exposed to chitin, a major component of the *C. neoformans* cell wall (46–50). Previous studies showed that host CLPs contribute

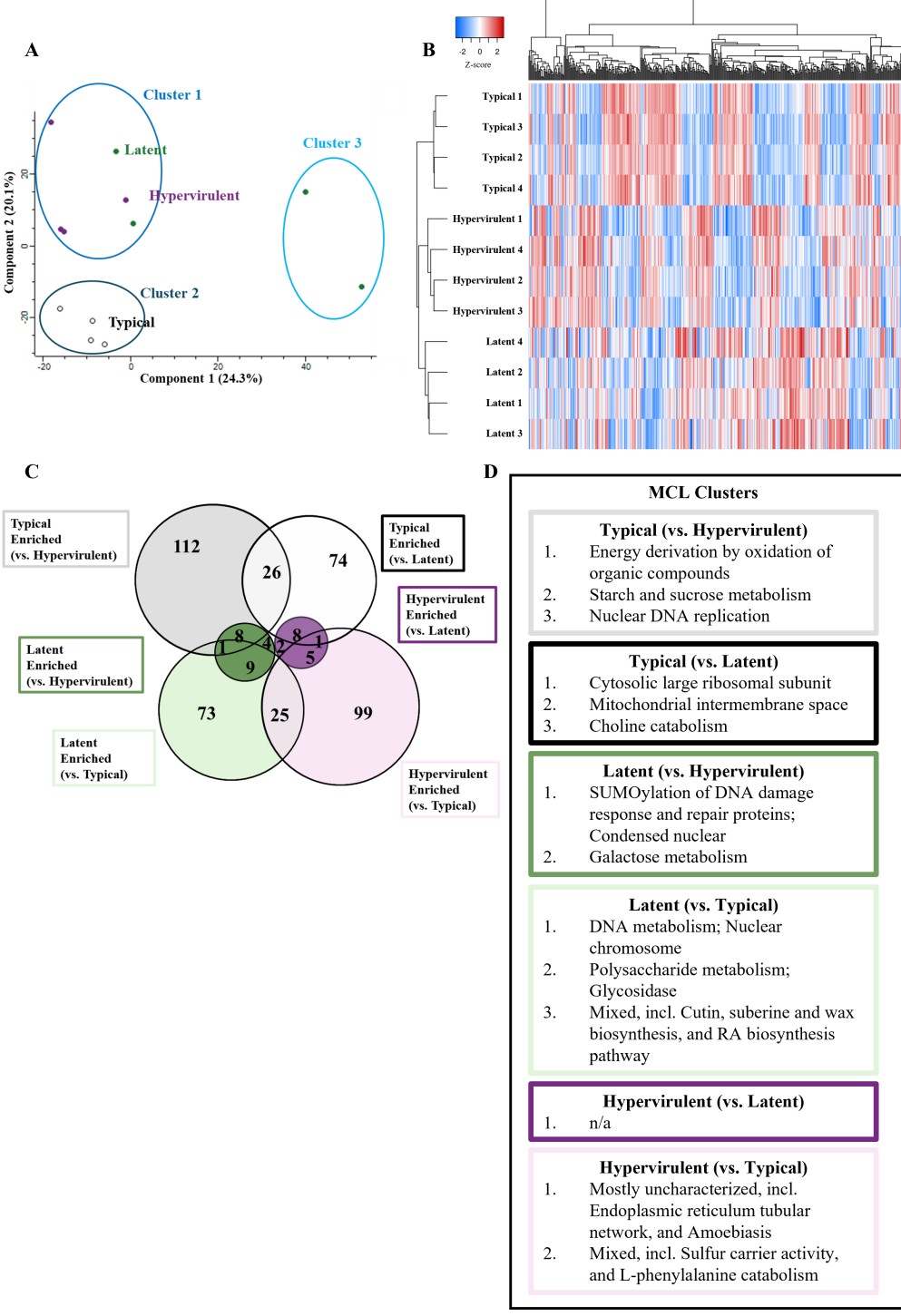

**FIG 7** *Cryptococcus* clinical isolates have similar *in vitro* proteomes. *C. neoformans* strains were grown in culture, and their proteomes were analyzed via LC-MS/MS to identify *in vitro* differences. (A) PCA plot showing the clustering of typical (black), latent (green), or hypervirulent (purple) *C. neoformans* culture proteomes. Three clusters were generated from a K-means clustering algorithm and a Chi-squared analysis (*P*-value = 0.00302). (B) Euclidean-clustered heat map of all significantly altered proteins with hierarchical dendrogram organization of samples and proteins. (C) Venn diagram depicting the number of overlapping or significant proteins determined from Student's *t*-test with multiple comparison correction of each strain to one another. (D) MCL clustering (inflation parameter (k) = 1.5) of protein annotations to identify enrichment of annotated terms.

to the type 2-dominant immune responses during KN99α typical lethal infection (43, 50). Indeed, our findings here show that our type 2-dominant lethal infections produced greater abundances of CLPs, which supports that CLPs play a role in host defense response that may contribute to the ineffective type 2 fungal response.

We also observed changes in extracellular matrix proteins. Typical infection had increases in Mmp12 and B4galt1, which are involved in the degradation of extracellular matrix and immune cell infiltration, respectively (32, 34). The hypervirulent infection exhibited greater Prtn3, which degrades extracellular matrix to facilitate neutrophil migration (51). The higher abundance of proteins involved in extracellular matrix degradation during lethal infection is consistent with the loss of lung structural integrity and influx of immune cells. In contrast, the latent infection had increased abundance of collagens and fibrinogens that are involved in structural integrity for the lung parenchyma and, importantly, lung granulomas (6, 7, 37, 52).

Moreover, our analysis of immune-related proteins involved in granuloma formation revealed that CYBB and Ncf1, which are involved in NADH oxidase–mediated reactive oxygen species (ROS) generation, as well as the neutrophil and monocyte alarmin S100a9, are significantly lower in latent infection than in the lethal infections. Previous studies showed that deficiencies in CYBB and Ncf1 and increases in S100a9 promote granuloma formation (35, 37–39). Therefore, granuloma formation in latent infection, at least during the early phase, is likely attributed to deficiencies in ROS fungal killing. These differences in granuloma-forming immune proteins further underpin the significant deviation in granuloma formation capability between latent and lethal infection and the differences in lung tissue destruction that contribute to mortality (6, 52).

By analyzing the global proteome of *C. neoformans* cultures *in vitro*, our goal was to identify disease manifestation-specific differences in key virulence factors that may predispose strains to particular disease manifestations. We report here that in the tested strains, there are no differences in previously established virulence factors, such as chitin deacetylases, mannoproteins, urease, phospholipase, or melanin (53, 54). Instead, we found that our clinically derived strains were more similar to each other than to the laboratory-generated KN99α reference strain. This was unsurprising considering that both UgCl223 and UgCl422 were isolated from infected Ugandan patients with HIV (18) and are evolutionarily more similar, while KN99α is a laboratory-generated strain derived from the clinically isolated H99 strain collected in 1978 (55) from an American patient with Hodgkin's lymphoma and is evolutionarily divergent from the Ugandan isolates. Instead, we noted differences in sugar metabolism, DNA metabolism, and biosynthetic annotations between strains. Our interpretation, therefore, is that disease manifestation-specific differences between these strains arise from divergent stress response pathways rather than intrinsic differences in virulence factors.

Our analysis of the *C. neoformans* proteome post-infection revealed *in vivo* differences that further highlighted the contrast between latent and lethal disease. Firstly, our finding that the latent strain produced greater abundance of histones is surprising. While chromatin remodeling and histone acetylation were previously shown to be important for *C. neoformans* survival during host infection (56–58), our findings suggest that increased histone accumulation, either from deficient degradation pathways or increased production, might contribute to the survivability of this latent strain. Indeed, a study by Feser et al. showed that increasing the accumulation of histones in yeast increased lifespan and survivability (59). Histone deacetylase 1 (HDAC1) expression significantly impacts *C. neoformans* survival and virulence factor production (57). Furthermore, differential methylation of H3 may be biologically relevant as our latent strain was deficient in S-adenosylmethionine synthetase which synthesizes the universal methyl donor S-adenosylmethionine (SAM).

Interestingly, we found that latent and lethal *C. neoformans* strains differed in the abundance of two key stress response proteins, calmodulin and Hsp90-like protein, respectively. Calmodulin interacts with calcium ($Ca^{2+}$) and calcineurin, a serine/threonine-protein phosphatase which was also elevated in our latent strain, to promote

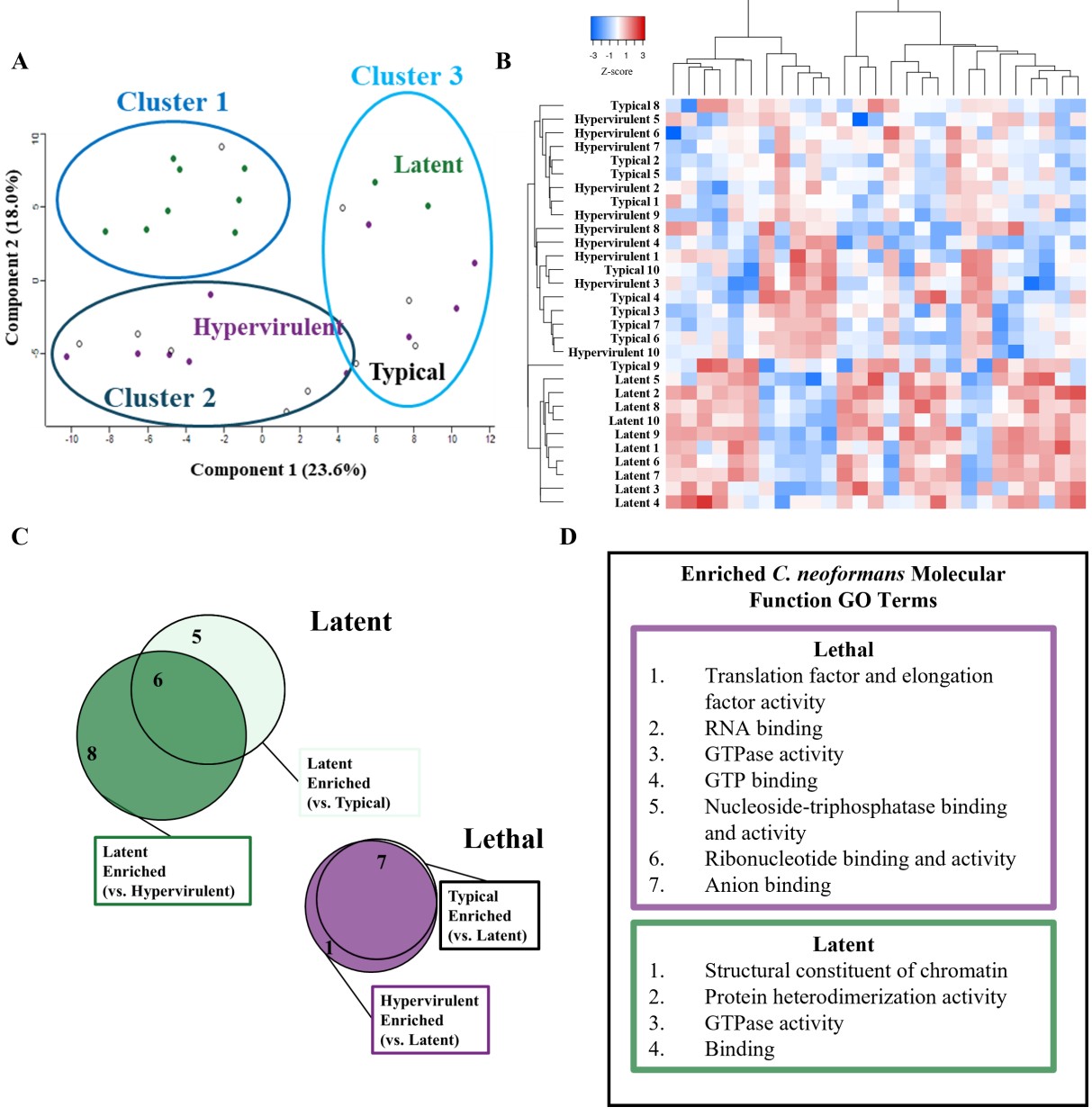

**FIG 8** Latent disease-causing UgCl223 produces a significantly different *in vivo* proteome compared to lethal disease-causing KN99α (typical) and UgCl422 (hypervirulent). Global proteomic analysis of the typical (KN99α), latent (UgCl223), or hypervirulent (UgCl422) *C. neoformans* strains from infected mouse lungs at 14 days post-infection. (A) PCA plot showing the clustering of typical (black), latent (green), or hypervirulent (purple) *C. neoformans* infection proteomes. Three clusters were generated from a K-means clustering algorithm and Chi-squared analysis (*P*-value = 0.000750). (B) Euclidean-clustered heat map of all significantly altered proteins with hierarchical dendrogram organization of samples and proteins. (C) Venn diagram depicting the number of overlapping or significant proteins determined from Student's *t*-test with multiple comparison correction of the strain to each other. The typical and hypervirulent strains were not significantly different from each other and were grouped as "lethal," and all the latent enriched proteins were grouped together as "latent." (D) STRING pathway analysis showing the significantly enriched molecular function gene ontology (GO) terms for lethal (purple) or latent (green) post-infection strains.

thermotolerance mechanisms and *in vivo* survival (60). In contrast, our typical and hypervirulent lethal strains produced greater Hsp90-like protein. Hsp90 has been shown to promote *C. neoformans* thermotolerance and survival *in vivo* as well, but through a still unknown mechanism (61–63). Utilizing a copper-repressible system, Fu et al. demonstrated that depleting Hsp90 transcripts during *in vivo* lung infection attenuated virulence (61), while Chatterjee et al. showed that fluorescent Hsp90 localized to the *C.*

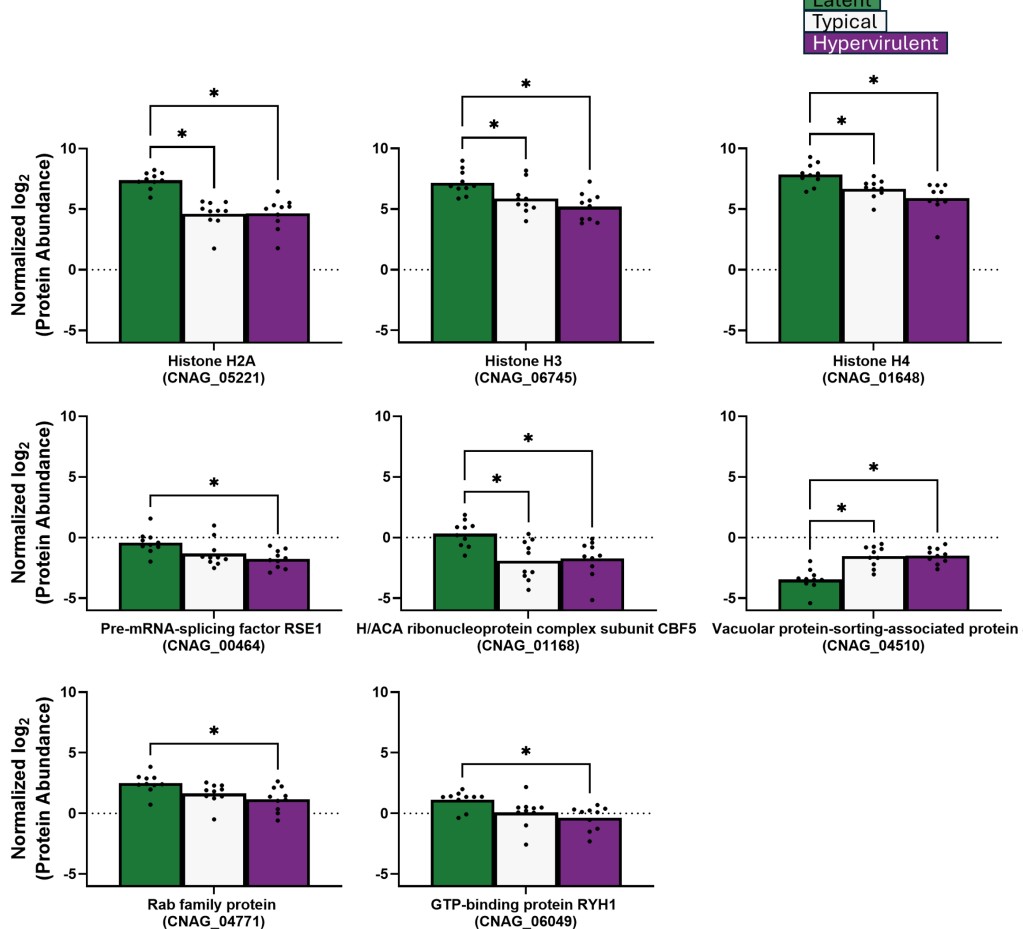

**FIG 9** Latent *C. neoformans* strain produces more nucleosome core-related proteins than lethal strains *in vivo*. Abundances of *C. neoformans* proteins related to nucleosome core function were analyzed by one-way ANOVA with Tukey's post-hoc correction. *$P$-value <0.05.

*neoformans* cell surface (63). We show here that Hsp90-like protein is associated with virulent *C. neoformans* disease manifestations. Therefore, Hsp90 may be an antigen of interest during cryptococcal infection, similar to how Hsp90 is a major component of a quadrivalent tuberculosis vaccine (64).

It is important to note that we detected substantially fewer total cryptococcal proteins from infected lungs than *in vitro* culture (116 from lungs versus 4,413 from cultures). Our observation of reduced *C. neoformans* protein identifications in infected mouse lungs likely highlights the need for improvement in the experimental methodology for dual perspective Cryptococcus proteomics. Here, we homogenized lung tissue using bead beating, sonication, and chemical lysis methods, which may have produced greater amounts of mouse proteins compared to *C. neoformans* proteins leading to lower detection of fungal peptides. Alternatively, the fungal cells may not have been adequately lysed due to their location within the lung tissue. Additionally, we were not able to detect the major antigenic proteins that exist on the fungal capsule and cell wall, such as mannoproteins and chitin deacetylases, using our *in vivo* method. To improve the detection of capsule- and cell wall-associated proteins from infected tissue, *C. neoformans* cells could be isolated from host cells or tissue to enrich the sample for fungal components. However, this approach may also contribute to the loss of fungal cells, which can be problematic in infections with lower fungal burden, as is the case during latent infection. This discrepancy in proteome depth, especially for capsule- and

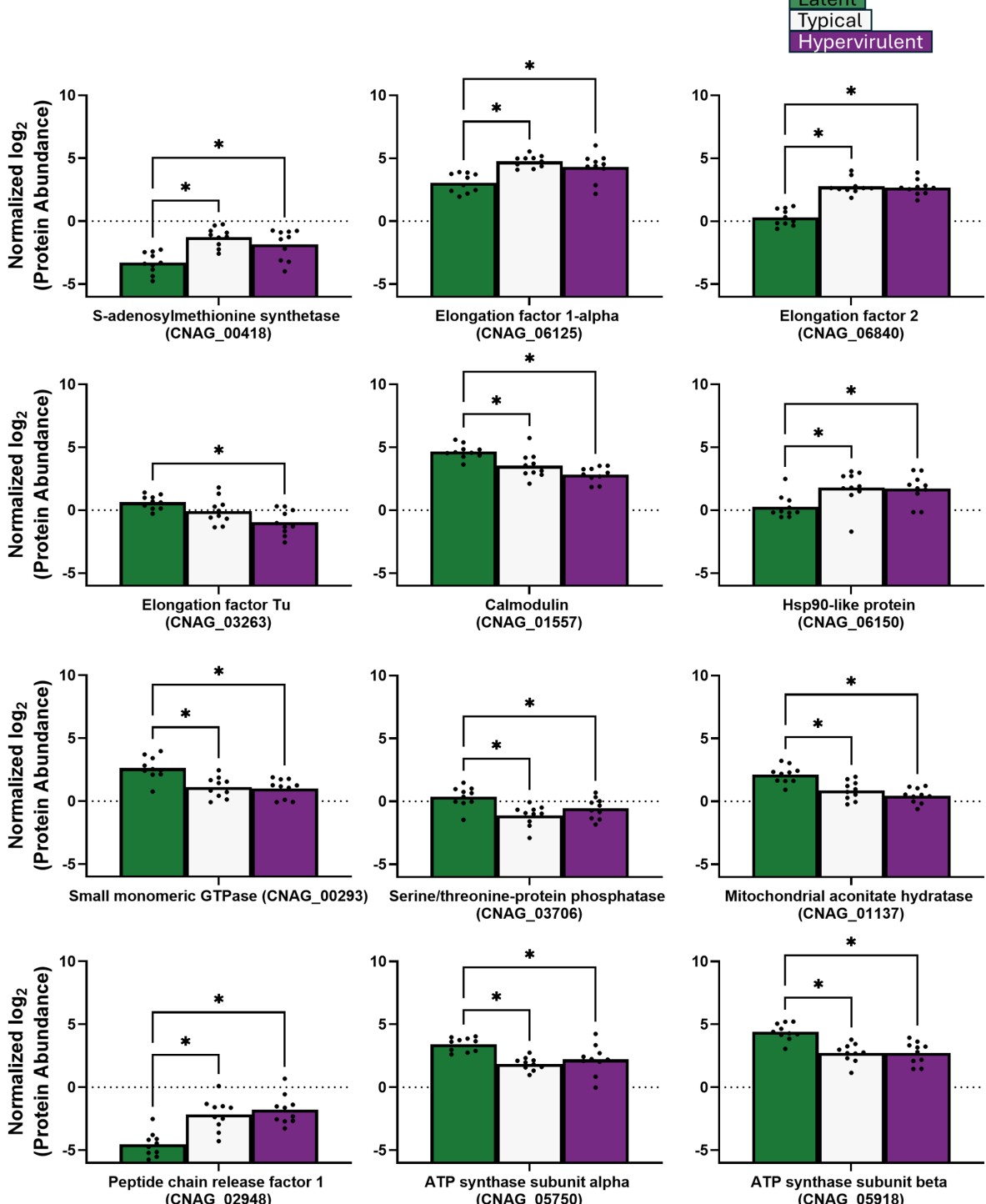

**FIG 10** Translation-related protein abundance varied across *C. neoformans* strains *in vivo*. Abundances of *C. neoformans* proteins involved in translation were analyzed by one-way ANOVA with Tukey's post-hoc correction. *P-value <0.05.

cell wall-associated proteins, underscores the current limitations of detecting both host and pathogen proteins from the same sample and should inform the next generation of proteomic analysis techniques.

We chose the 14 days post-infection (DPI) time point for all clinical isolate infections to capture the lung proteome before the mice began succumbing to disease. The

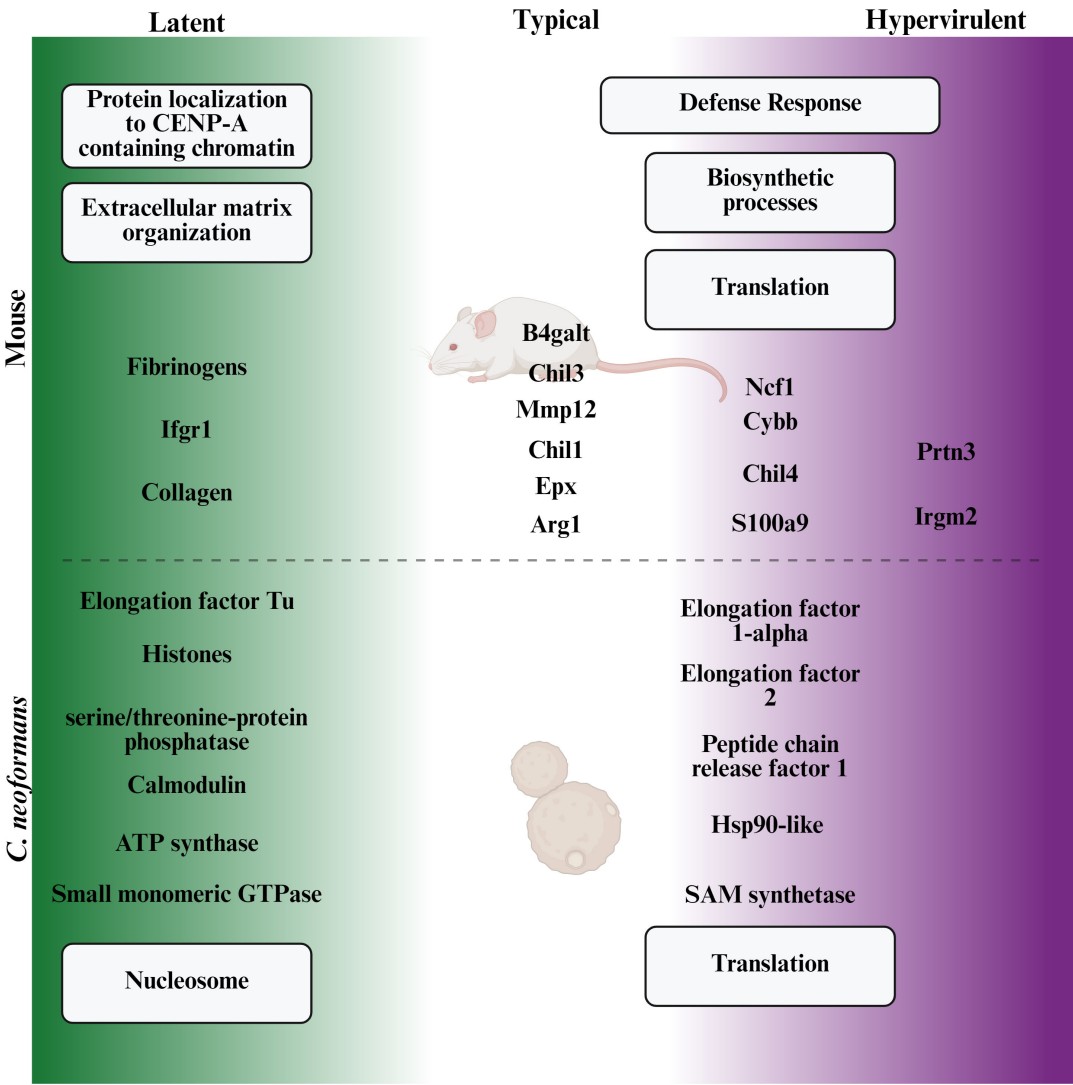

**FIG 11** Virulence-specific proteomic differences associated with different disease outcomes during cryptococcosis. Proteins (text not in boxes) and protein pathways (text in boxes) associated with the host-pathogen disease manifestation spectrum. In both the host and pathogen proteomes, latent disease is substantially different from lethal disease. In the 14 days post-infection mouse host, latent strain infection induces host pathways involved in extracellular matrix organization and chromatin interaction, while infection with typical or hypervirulent strains induces defense responses and translational pathways. A typical infection can be differentiated by the increased expression of Arg1, Epx, Mmp12, Chil1, Chil3, and B4galt, while hypervirulent infection increases Irgm2 and Prtn3. In *C. neoformans*, the latent strain exhibited enriched nucleosome histone proteins, ATP synthase, elongation factor Tu, small monomeric GTPase, serine/threonine-protein phosphatase, and the stress-response protein calmodulin. The proteomes of typical and hypervirulent isolates were not significantly different and both expressed increased translational proteins such as elongation factors 1-alpha and 2, peptide chain release factor 1, and the critical survival proteins SAM synthetase and Hsp90-like protein. Created in https://BioRender.com.

median survival for hypervirulent infections was defined to be at least two days less than the typical KN99α or around 15 DPI (3). To minimize the chance that mice with infections using hypervirulent strains would perish prior to collection, 14 DPI was chosen and standardized for all infections. It is likely that choosing a different time point would yield different proteomes for both the host and pathogen, especially in cases of the survivable latent infection (6, 7). While none of the mice exhibited noticeable symptoms of disease at 14 DPI, previous work has shown that mice infected with typical and hypervirulent isolates exhibit severe pulmonary damage (3). Moreover, the host response to cryptococcal infection changes over time (6), further supporting the idea that the proteomic profile of the host-pathogen is impacted by infection duration. Our findings

here are the first step of evaluating the proteomic profile of clinically relevant models of cryptococcal infection, and future studies should investigate other time points to determine how the proteomics of infection change.

Overall, we showed that the host proteomic responses to cryptococcal infection are dependent on disease manifestation-specific differences between strains and identified several novel proteins and pathway targets associated with these differences, which can be clinically and scientifically examined for diagnostic and research advancement. We also revealed disease manifestation-specific cryptococcal protein targets that should be further explored to determine their biological and clinical relevance during lung infection.

## ACKNOWLEDGMENTS

We thank the Mass Spectrometry Facility (MSF) at the University of Guelph for technical assistance with the LC-MS/MS workflow.

This work was supported by the National Institutes of Health R01AI176922 and R01NS118538 to K.N., and Canadian Institutes of Health Research Project Grants and the Canada Research Chairs program to J.G.M. J.B. was supported by the National Institutes of Health F31AI181528 and the University of Minnesota Medical Student Training Program T32GM008244. J.M.P. was supported by the University of Minnesota Targets of Cancer Predoctoral Training Program T32CA009138.

J.J.B., J.G.M., and K.N. conceived and designed the experiments. J.J.B. and J.M. performed the experiments. J.J.B., J.M.P., J.G.M., J.M., S.T., and K.N. analyzed the data. D.B.M., J.G.M., and K.N. contributed reagents, materials, and/or analysis tools. J.J.B. and K.N. wrote the paper. All authors contributed to the article and approved the submitted version.

## AUTHOR AFFILIATIONS

[1]Department of Microbiology and Immunology, University of Minnesota, Minneapolis, Minnesota, USA
[2]Microbiology, Immunology, and Cancer Biology Graduate Program, University of Minnesota, Minneapolis, Minnesota, USA
[3]Department of Molecular and Cellular Biology, University of Guelph, Guelph, Ontario, Canada
[4]Department of Laboratory Medicine and Pathology, University of Minnesota, Minneapolis, Minnesota, USA
[5]Infectious Diseases Institute, Makerere University, Kampala, Uganda
[6]Department of Biomedical Sciences and Pathobiology, Virginia Tech University, Blacksburg, Virginia, USA

## AUTHOR ORCIDs

Jovany J. Betancourt http://orcid.org/0000-0003-0364-4816
Jennifer Geddes-McAlister http://orcid.org/0000-0002-4257-2096
Kirsten Nielsen http://orcid.org/0000-0002-2318-168X

## AUTHOR CONTRIBUTIONS

Jason A. McAlister, Formal analysis, Investigation, Methodology, Resources, Supervision, Writing – review and editing | Jesenia M. Perez, Formal analysis, Methodology, Visualization, Writing – review and editing | David B. Meya, Resources | Stefani N. Thomas, Resources, Supervision, Validation, Writing – review and editing | Jennifer Geddes-McAlister, Conceptualization, Funding acquisition, Investigation, Methodology, Project administration, Resources, Supervision, Writing – review and editing | Kirsten Nielsen, Conceptualization, Funding acquisition, Investigation, Methodology, Project administration, Resources, Supervision, Writing – review and editing.

## DATA AVAILABILITY

The mass spectrometry proteomics data have been deposited to the ProteomeXchange Consortium via the PRIDE (21) partner repository with the dataset identifier PXD063417 (Project accession: PXD063417, Token: 9DZ3u5h8R6g7).

## ETHICS APPROVAL

The animal study was reviewed and approved by the University of Minnesota Institutional Animal Care and Use Committee.

## ADDITIONAL FILES

The following material is available online.

### Supplemental Material

**Supplemental Material (mSystems00751-25-s0001.docx).** Figures S1 to S3 and captions for supplemental tables.
**Supplemental Tables (mSystems00751-25-s0002.pdf).** Tables S1 to S3.

### Open Peer Review

**PEER REVIEW HISTORY (review-history.pdf).** An accounting of the reviewer comments and feedback.

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
