## [Reviewer comments · mSystems]

Global proteomic analysis of *Cryptococcus neoformans* clinical strains reveals significant differences between latent and lethal infection

Jovany Betancourt, Jason McAlister, Jesenia Perez, David Meya, Stefani Thomas, Jennifer Geddes-McAlister, and Kirsten Nielsen

Corresponding Author(s): Kirsten Nielsen, Virginia-Maryland College of Veterinary Medicine

Review Timeline:

Submission Date:	June 2, 2025
Editorial Decision:	August 22, 2025
Revision Received:	September 2, 2025
Accepted:	September 3, 2025

Editor: Laura-Isobel McCall

Reviewer(s): Disclosure of reviewer identity is with reference to reviewer comments included in decision letter(s). The following individuals involved in review of your submission have agreed to reveal their identity: Camaron R Hole (Reviewer #1)

Transaction Report:

DOI: <https://doi.org/10.1128/msystems.00751-25>

Re: mSystems00751-25 (Global proteomic analysis of *Cryptococcus neoformans* clinical strains reveals significant differences between latent and lethal infection)

Dear Dr. Kirsten Nielsen:

Overall, reviewers appreciated the value of the study. Only a few issues were noted that can all be addressed with text modifications.

Revision Guidelines

Sincerely,
Laura-Isobel McCall
Editor
mSystems

Reviewer #1 (Comments for the Author):

This manuscript presents a comprehensive global proteomic analysis of mouse lung tissue and *Cryptococcus neoformans* clinical isolates to investigate the host and pathogen factors underlying latent versus lethal cryptococcal infections. Using in vivo infection models and mass spectrometry-based proteomics, the authors demonstrate that latent infection elicits a distinct host

response characterized by extracellular matrix remodeling and granuloma-associated pathways, while lethal infections, caused by both typical and hypervirulent strains, induce immune defense, translational, and metabolic responses. The experiments are rigorous and well-controlled. The use of both latent and lethal clinical isolates provides valuable insights into the divergent host-pathogen dynamics that underlie different disease manifestations. The dual-perspective proteomics approach is innovative and informative, allowing parallel insight into both host and pathogen responses. The work is highly relevant to the field of cryptococcal pathogenesis and host immune response.

Minor Concerns and Recommendations:

1. While the Materials and Methods section clearly states that mice were euthanized at 14 days post-infection, this detail is missing from the Results section and figure legends.
2. There are occasional grammatical inconsistencies and awkward phrasings that slightly detract from the manuscript's clarity. Examples include Lines 71-73, 80-83, and 214-218

Reviewer #2 (Comments for the Author):

This study investigated the proteome profiles of both the host (A/J mice) and the pathogen (different strains of *C. neoformans*) during intranasal-induced infections. Overall, the manuscript was clearly written and easy to follow. I have only a few comments for authors' consideration and clarification.

1. Please present or cite evidence that A/J mouse is an appropriate model for human infection and disease by *C. neoformans*.
2. Please present or cite evidence that strain UgC1223 causes latent infection. While ref. #3 presented evidence for strains KN99alpha and UgC1422 as representatives of typical lethal and hypervirulent strains respectively, I couldn't find information on strain UgC1223 in that study.
3. Why was day 14 chosen for euthanization? Would a different time point further down show different proteome profiles? What symptoms (e.g., morphological and behavioural) were present in various groups of infected mice at that time? It would be informative to present the above in order to link proteome changes and disease progression.

Reviewer #3 (Comments for the Author):

This is a well-written and comprehensive study that demonstrates advanced proteomic profiling to dissect both host and fungal responses during *Cryptococcus neoformans* infection. The authors compared proteomes of latent, typical, and hypervirulent clinically isolated *C. neoformans* strains in murine models and provide valuable insight into specific proteomic signatures associated with infection. The dual analysis (both host and pathogen proteomes) is a major strength, and the dataset represents a valuable resource for the field.

Just a few minor comments are:

1. While the study provides a global analysis of the host and pathogen proteomes in vivo, the number of cryptococcal proteins detected from infected tissue was substantially reduced compared to in vitro cultures (116 vs. 4413 proteins). I wonder whether the proteomic sample preparation steps (e.g., bead beating or sonication) may have differentially impacted the recovery of host versus fungal proteins. Please clarify this point.
2. The manuscript notes that capsule- and cell wall-associated proteins were not recovered from lung proteomes. It would be more helpful to readers if the authors could expand the discussion on potential methodological strategies to overcome this limitation in future studies.

Global proteomic analysis of *Cryptococcus neoformans* clinical strains reveals significant differences between latent and lethal infection

Reviewer Responses

Reviewer 1

We thank Reviewer 1 for their constructive edits and suggestions. Here are our responses:

Comment: “While the Materials and Methods section clearly states that mice were euthanized at 14 days post-infection, this detail is missing from the Results section and figure legends.”

Response: Added “at 14 days post-infection” in the results section and figure legends where appropriate.

Comment: “There are occasional grammatical inconsistencies and awkward phrasings that slightly detract from the manuscript's clarity. Examples include Lines 71-73, 80-83, and 214-218”

Response: Highlighted sentences were rephrased for clarity, and the rest of the manuscript was double-checked manually and using Grammarly software.

Reviewer 2

We thank Reviewer 2 for their constructive edits and suggestions. Here are our responses:

Comment: “Please present or cite evidence that A/J mouse is an appropriate model for human infection and disease by *C. neoformans*.”

Response: Our previous studies showed that the inhalation model using A/J mice recapitulates the virulence observed during human disease, with clinical isolates from patients who died showing rapid mortality in the A/J mouse model and isolates from patients that survived their infections showing attenuated virulence in the A/J mouse model (Ref. 5). Thus, this A/J mouse model is the model that should be used when studies aim to draw correlations between human disease. Citations (Ref. 3 & 5) that used A/J mice for clinically relevant *C. neoformans* infection modeling were added to the Mouse infections section of the Materials and Methods.

Comment: “Please present or cite evidence that strain UgCl223 causes latent infection. While ref. #3 presented evidence for strains KN99alpha and UgCl422 as representatives of typical lethal and hypervirulent strains respectively, I couldn't find information on strain UgCl223 in that study.”

Response: Citation (Ref. 7) showing the use of UgCl223 to establish latent infection was added to the Mouse Infections and Results sections.

Comment: “Why was day 14 chosen for euthanization? Would a different time point further down show different proteome profiles? What symptoms (e.g., morphological and behavioural) were present in various groups of infected mice at that time? It would be informative to present the above in order to link proteome changes and disease progression.”

Response: We thank the reviewer for this helpful comment. We added the following statement to the Discussion section: “We chose the 14 days post-infection (DPI) timepoint for all clinical isolate infections to capture the lung proteome before mice began succumbing to disease. The median survival for hypervirulent infections was defined to be at least two days less than the typical KN99 α , or around 15 DPI (Jackson et al., 2024). To minimize the chance that mice with infections using hypervirulent strains would perish prior to collection, 14 DPI was chosen and standardized for all infections. It is likely that choosing a different timepoint would yield different proteomes for both the host and pathogen, especially for the latent infection. While none of the mice exhibited noticeable symptoms of disease at 14 DPI, previous work has shown that mice infected with typical and hypervirulent isolates exhibit severe pulmonary damage. Moreover, the host response to cryptococcal infection changes over time (Betancourt et al., 2025), further supporting the idea that the proteomic profile of the host-pathogen is impacted by infection duration. Our findings here are the first step of evaluating the proteomic profile of clinically relevant models of cryptococcal infection, and future studies should investigate other timepoints to determine how the proteomics of infection changes.”

Reviewer 3

We thank Reviewer 3 for their constructive edits and suggestions. Here are our responses:

Comment: “While the study provides a global analysis of the host and pathogen proteomes *in vivo*, the number of cryptococcal proteins detected from infected tissue was substantially reduced compared to *in vitro* cultures (116 vs. 4413 proteins). I wonder whether the proteomic sample preparation steps (e.g., bead beating or sonication) may have differentially impacted the recovery of host versus fungal proteins. Please clarify this point.”

AND

Comment: “The manuscript notes that capsule- and cell wall-associated proteins were not recovered from lung proteomes. It would be more helpful to readers if the authors could expand the discussion on potential methodological strategies to overcome this limitation in future studies.”

Response: We thank the reviewer for these helpful comments on a topic that we feel is important for readers to fully understand about our studies. Additional details were added to the referenced Discussion section paragraph as follows:

“It is important to note that we detected substantially fewer total cryptococcal proteins from infected lungs than *in vitro* culture (116 from lungs versus 4413 from cultures). Our observation of the reduced *C. neoformans* protein identifications from infected mouse lungs likely highlights the need for improvement in the experimental methodology for dual perspective Cryptococcus proteomics. Here, we homogenized lung tissue using bead beating, sonication, and chemical lysis methods which may have produced greater amounts of mouse proteins compared to *C. neoformans* proteins leading to lower detection of fungal peptides. Alternatively, the fungal cells may not have been adequately lysed due to their location within the lung tissue. Additionally, we were not able to detect the major antigenic proteins that exist on the fungal capsule and cell wall such as mannoproteins and chitin deacetylases from our *in vivo* method. To improve the detection of capsule- and cell wall-associated proteins from infected tissue, *C. neoformans* cells could be isolated from the host cells or tissue to enrich the sample for fungal components. However, this approach may also contribute to loss of fungal cells which can be problematic in infections with lower fungal burden, as is the case during latent infection. This discrepancy in proteome depth, especially for capsule- and cell wall-associated proteins, underscores the current limitations of detecting both host and pathogen proteins from the same sample and should inform the next generation of proteomic analysis techniques.

Re: mSystems00751-25R1 (Global proteomic analysis of *Cryptococcus neoformans* clinical strains reveals significant differences between latent and lethal infection)

Dear Dr. Kirsten Nielsen:

Your manuscript has been accepted, and I am forwarding it to the ASM production staff for publication. Your paper will first be checked to make sure all elements meet the technical requirements. ASM staff will contact you if anything needs to be revised before copyediting and production can begin. Otherwise, you will be notified when your proofs are ready to be viewed.

Cover Image Submissions: If you would like to submit a potential Cover Image, please email a file and a short legend to mSystems@asmusa.org. Please note that we can only consider images that (i) the authors created or own and (ii) have not been previously published. By submitting, you agree that the image can be used under the same terms as the published article. Image File requirements: TIF/EPS, 7.5 inches wide by 8.25 inches tall (at least 2,250 pixels wide by 2,475 pixels tall), minimum 300 dpi resolution (600 dpi preferred), RGB, and no figure elements, e.g., arrows or panel labels. The legend should be a short description of the image, 1-2 sentences recommended. Please download and use this interactive template in Adobe to ensure that your proposed cover image meets our size requirements (<https://journals.asm.org/pb-assets/pdf-text-excel-files/ASM-Interactive-Sizing-Cover-Template-1715689791.pdf>).

Sincerely,
Laura-Isobel McCall
Editor
mSystems